# Lithium, Inflammation and Neuroinflammation with Emphasis on Bipolar Disorder—A Narrative Review

**DOI:** 10.3390/ijms252413277

**Published:** 2024-12-11

**Authors:** Odeya Damri, Galila Agam

**Affiliations:** Department of Clinical Biochemistry and Pharmacology, Faculty of Health Sciences, Zlotowski Center for Neuroscience and Zelman Center—The School of Brain Sciences and Cognition, Ben-Gurion University of the Negev, Beer-Sheva 8410501, Israel; odeyad@post.bgu.ac.il

**Keywords:** lithium, neuroinflammation, animal models, clinical trials

## Abstract

This narrative review examines lithium’s effects on immune function, inflammation and cell survival, particularly in bipolar disorder (BD) in in vitro studies, animal models and clinical studies. In vitro studies show that high lithium concentrations (5 mM, beyond the therapeutic window) reduce interleukin (IL)-1β production in monocytes and enhance T-lymphocyte resistance, suggesting a protective role against cell death. Lithium modulates oxidative stress in lipopolysaccharide (LPS)-activated macrophages by inhibiting nuclear factor (NF)-ƙB activity and reducing nitric oxide production. At therapeutically relevant levels, lithium increased both pro-inflammatory [interferon (INF)-γ, IL-8 and tumor necrosis factor (TNF)-α)] and anti-inflammatory (IL-10) cytokines on whole blood supernatant culture in healthy volunteers, influencing the balance of pro- and anti-inflammatory responses. Animal models reveal lithium’s potential to alleviate inflammatory diseases by reducing pro-inflammatory cytokines and enhancing anti-inflammatory responses. It also induces selective macrophage death in atherosclerotic plaques without harming other cells. In primary rat cerebellum cultures (ex vivo), lithium prevents neuronal loss and inhibits astroglial growth, impacting astrocytes and microglia. Clinical studies show that lithium alters cytokine profiles and reduces neuroinflammatory markers in BD patients. Chronic treatment decreases IL-2, IL-6, IL-10 and IFN-γ secretion from peripheral blood leukocytes. Lithium response correlates with TNF-α levels, with poor responders showing higher TNF-α. Overall, these findings elucidate lithium’s diverse mechanisms in modulating immune responses, reducing inflammation and promoting cell survival, with significant implications for managing BD and other inflammation-related conditions. Yet, to better understand the drug’s impact in BD and other inflammatory/neuroinflammatory conditions, further research is warranted to appreciate lithium’s therapeutic potential and its role in immune regulation.

## 1. Introduction

Lithium salts (lithium) have been the prototype drug for bipolar disorder (BD) for more than seven decades, exerting its therapeutic beneficial effects at a narrow window of plasma concentrations [1]. About 30% of lithium-treated patients achieve a complete response and another 33% experience a partial response [2,3,4,5,6,7].

Lithium’s mechanism of action has not yet been fully unraveled but data show that it involves multifaceted interactions within the central nervous system. Among other described effects, it inhibits the enzyme inositol monophosphatase 1 (IMPA1) in its therapeutic levels [8,9], leading to decreased levels in the two second messengers, inositol trisphosphate (IP3) and diacylglycerol (DAG) [10,11,12]. This disruption in inositol signaling cascades can influence neurotransmission, cellular signaling, synaptic plasticity and autophagy [10,13,14,15], possibly contributing to its mood-stabilizing effects [16,17]. In addition, lithium has garnered attention for its potential anti-inflammatory effects. Another target suggested as lithium’s mechanism of action is the enzyme glycogen synthase kinase-3 (GSK-3), also claimed to be lithium-inhibitable, impacting various intracellular signaling pathways implicated in mood regulation [11,15,18,19]. This inhibition is claimed to modulate neurotransmitter signaling, particularly involving serotonin and dopamine, which are crucial for mood stabilization [20,21], and to dampen inflammatory signaling pathways [22]. Hence, while the exact mechanism is still being elucidated, research suggests that lithium may modulate various components of the immune system, leading to decreased inflammation [22,23,24,25,26,27,28]. If lithium`s modulation of inflammation is involved in its mood stabilizing effect it may (1) shed light on the pathophysiology of the disease and (2) serve as a rationale platform for the search/design of potential lithium mimetics/novel anti-inflammatory drugs.

Inflammation is often determined by measuring pro- and anti-inflammatory cytokines. As for BD, a comprehensive review of 102 papers reporting inflammatory biomarkers including interleukins, interferon (INF)γ, tumor necrosis factor (TNF)α, transforming growth factor (TGF)β1 and chemokines levels in serum/plasma, cerebrospinal fluid (CSF), a variety of peripheral white blood cells (PBLs) and postmortem brain summarized that pro-inflammatory cytokine are elevated and anti-inflammatory cytokines are reduced in patients afflicted with this disorder [29]. In general, research indicates that both peripheral and central inflammation is strongly associated with, and is potentially a driving factor in, neuropsychiatric diseases [30,31,32], including mood disorders such as depression [33,34,35] and BD [35,36,37]. Further, many of the therapeutics used to treat these diseases, including lithium, can also impact inflammation [38,39].

Lithium has been found to influence the production of pro-inflammatory cytokines, such as interleukin (IL)-6 and TNF-α, while enhancing the secretion of anti-inflammatory cytokines like IL-10 [23]. The effects of cytokines in the central nervous system (CNS) include an array of metabolic, endocrine hypothalamic–pituitary–adrenal (HPA) axis and behavioral alterations [40,41,42]. In bipolar patients, during both mania and depression episodes, elevated pro-inflammatory cytokines levels are consistently reported while reports related to anti-inflammatory cytokines are inconsistent [37]. Among pro-inflammatory cytokines reports, IL-1β primarily relates to the depressive state [43]. Lithium has been reported to affect immune modulation via second messengers’ systems and transcription factors [44,45,46].

In the present narrative review, we summarize lithium’s anti-inflammatory effects in vitro, in vivo and in clinical trials.

## 2. Methodology of the Literature Search

PubMed was searched for the years 1990–2024 using the key words “lithium and inflammation or neuroinflammation and bipolar disorder”, including only article types of either case reports; classical articles; clinical studies; clinical trials—phase II, phase III and phase IV; veterinary; meta-analysis; and multicenter studies, followed by integration according to the headlines as follows.

## 3. Lithium’s In Vitro Anti-Inflammatory Effects

In this section, we reviewed eight papers that mostly indicated the anti-inflammatory effects of lithium. Given that therapeutically relevant blood lithium levels are in the range of 0.6 to 1.6 mM, we denote whenever the concentration used is higher, since levels exceeding this range in patients are toxic and even detrimental.

In vitro exposure of monocytes to high concentrations of lithium chloride (5 mM, beyond the therapeutic window of the drug) resulted in decreased production in IL-1β without affecting IL-6 production [47]. The study by Pietruczuk et al. evaluated the effects of lithium on T-lymphocytes derived from patients with BD [48]. Results revealed that BD patients’ T-lymphocytes, particularly those treated with lithium, exhibited reduced proliferation capacity compared to cells derived from healthy individuals. In vitro lithium treatment of patients’ T-lymphocytes did not significantly influence the proliferation capacity but did demonstrate a dose-dependent protective effect against apoptosis of the cells, suggesting the drug’s potential in mitigating cell death in BD and highlighting the drug’s therapeutic potential in managing cell survival [48].

Another study aimed to elucidate lithium’s regulatory mechanisms on oxidative stress in lipopolysaccharide (LPS)-activated macrophages by assessing its impact on nuclear factor-ƙB (NF-ƙB) activity and on mRNA expression of oxidative stress-related NF-ƙB genes. Treatment of RAW 264.7 macrophages with lithium up to 10 mM did not affect cell proliferation, viability, growth or adhesion [49]. In a later study of the same group carried out to delineate the regulatory mechanism of lithium on oxidative stress using the same paradigm (LPS-activated RAW 264.7 macrophages), the authors reported that pretreatment with 1.25 mM lithium (a therapeutically relevant concentration) reduced nitric oxide (NO) production [50]. On the other hand, lithium modulated the expression of inflammatory genes, including inhibitor of nuclear factor (*I*)*κB-α*, tumor necrosis factor receptor associated factor (*TRAF*)*3*, toll-interacting protein (*Tollip*) and *NF-ƙB1/p50*, which regulate the NF-ƙB pathway, and inhibited NF-ƙB activity by reducing nuclear translocation in activated macrophages. This study was the first to link Tollip, TRAF-3 and IƙB-α mRNA expression with lithium’s effects on NF-ƙB activity [49]. All other factors, e.g., reactive oxygen species (ROS) production, the *IƙB-α*, *NF-ƙB1/p50* genes and NF-ƙB nuclear translocation, that were found to be normalized by lithium were assessed using only much higher lithium concentrations than its therapeutic range (10 mM), questioning the relevance of these changes to the drug’s in vivo effect on macrophages [51,52,53].

An in vitro study by Maes et al. [54] explored lithium’s in vitro effect at low/therapeutic concentrations on cytokine production in the unstimulated and phytohemagglutinin (PHA) + LPS-stimulated diluted whole blood of healthy volunteers cultured for 72 h. At therapeutic concentrations, lithium significantly increased the production of pro-inflammatory cytokines) INF-γ, IL-8 and TNF-α (and of negative immunoregulatory cytokines or proteins (IL-10 and interleukin-1 receptor antagonist), suggesting that lithium possesses significant immunoregulatory effects, potentially influencing the balance of pro-inflammatory and anti-inflammatory responses [54].

De Mayer et al. demonstrated that 30 mM lithium for 7 days reduces the presence of macrophages in atherosclerotic plaques, an effect achieved through the selective induction of macrophage death via inositol monophosphatase inhibition. Importantly, this effect occurs without compromising the viability/functionality of smooth muscle cells and endothelial cells [55].

In primary mixed cultures prepared from cell suspensions of mechanically dissociated cerebellum taken from rat pups, treatment with 5 mM lithium prevented neuronal loss and, at the same time, inhibited astroglial growth. Conceivably, the direct impact on astrocytes and microglia could potentially influence the effects of lithium on neurons [56].

Lithium’s suppression of LPS effects, including microglial activation, a significant reduction in pro-inflammatory factors production and an increase in anti-inflammatory factors obtained by 1–2 mM, are robustly described, yet different mediators—toll-like receptor 4 expression inhibition or phosphoinositide 3-kinase (PI3K)/protein kinase B (PKB or Akt)/forkhead box O (FoxO)1 pathway activation via the PI3K/Akt/NF-ƙB cascade—have been reported [25,57,58,59].

Lithium’s anti-inflammatory effects were reflected in reduced secretion of the pro-inflammatory cytokines IL-6 and TNF-α in LPS-stimulated dendritic cells exposed to a concentration of 7.5 mM. Additionally, lithium significantly increased the production of the anti-inflammatory cytokine IL-10. These findings suggest that lithium may modulate inflammation by enhancing anti-inflammatory responses and could offer therapeutic potential for inflammation-related diseases [60].

## 4. Lithium’s Anti-Inflammatory Effects in Animal Models

A seminal review by Nassar and Azab [23] ten years ago eloquently summarized the in vivo effects of lithium treatment on inflammatory-related parameters. In short, in LPS-treated mice, the drug decreased TNF-α production and inhibited IL-6 production in plasma and the brain; reduced INF-γ production in serum and multiple organs of bacteria-treated mice; increased proliferation of oligodendrocytes, which led to enhanced myelination of optic nerves; and decreased brain microglia activation and cyclooxygenase (COX)-2 expression after a traumatic brain injury.

Sudduth et al. reported that 0.2% lithium in the diet of APPSwDI/NOS22/2 mice (an Alzheimer’s-like model) for a period of eight months significantly altered the neuroinflammatory phenotype of the brain. Namely, it significantly lowered the expression levels of *IL-1β*, *IL-6*, the macrophage receptor with a collagenous structure (*MARCO*), *TNFα* and *TNFα* receptor 1 (*TNFaR1*), which were elevated in the model mice as compared with wildtype mice [61].

Leu et al. reported that in mice, lithium treatment effectively alleviates the severity of collagen-induced arthritis triggered by LPS [60].

Another study reported that inflammation- and chemotaxis-relevant genes were significantly over-represented among Li-induced genes in monocyte-derived dendritic cells, and that GSK-3 plays a role in the regulation of complement (C)3 production [62], an essential protein in the complement system [63].

In a lisdexamfetamine-induced mania-like model in rats, lithium prevented the increase in inducible nitric oxide synthase (iNOS) levels but did not affect the increase in the inflammatory markers IL-1β, IL-10 and TNF-α following exposure to immune activation (LPS) [64].

A recent study in Wistar Albino rats treated with increasing doses of lithium found reduced counts of peripheral blood lymphocytes anti-inflammatory immune cells [helper T cells (CD4+), cytotoxic T cells (CD8+) and natural killer cells (CD161+)], whereas the drug’s effect on serum levels of pro-inflammatory cytokine levels were equivocal—some elevated (TNF-α and IFN-γ), others reduced (IL-1β and IL-2), or unchanged (IL-6). Concerning anti-inflammatory cytokines, IL-4 levels were reduced with no change in IL-10 [65].

## 5. Lithium’s Anti-Inflammatory Effects in Clinical Trials

Most of the clinical trials showed a strong association between lithium treatment and cytokines and neuroinflammatory markers, mainly in the serum and PBLs [26,27,66]. Patients under chronic lithium treatment or patients never medicated before were reported after 3 months of lithium treatment to have (ex vivo) significantly lower IL-2-, IL-6-, IL-10- and IFN-γ-secreting PBLs compared to healthy volunteers, but in vitro lithium treatment of PBLs did not affect the number of cytokine secreting cells derived from either patients or from healthy volunteers [27]. Another study reported that soluble serum IL-2 receptors (SIL-2RS) and SIL-6RS were increased in symptomatic rapid cycling bipolar patients, suggesting mild immune activation, which was normalized following lithium treatment. Yet, in normal volunteers, lithium treatment increased (rather than decreased) IL-2, SIL-2RS and SIL-6RS [26].

The specific changes in cytokines observed seem highly dependent on the disease phase and medication status, especially regarding lithium [26,27]. Guloksuz et al. explored the relationship between lithium response and TNF-α levels in bipolar disorder. Sixty euthymic BD patients on lithium therapy were evaluated for TNF-α levels, with response categorized using the ALDA lithium response scale (LRS) [66]. Patients with poor lithium response exhibited significantly higher TNF-α levels compared to good responders. Although those with partial response showed a similar trend, it was less pronounced. While the study acknowledges limitations, such as its retrospective nature, it suggests that TNF-α levels may play a role in lithium response and warrants further investigation into immune changes in treatment-resistant BD patients [66]. Isgren et al. analyzed cytokine levels in the CSF of 121 euthymic BD patients and 71 age- and sex-matched controls. The results indicated significantly higher CSF IL-8 levels in the BD patients. They also found that IL-8 concentrations associated positively both with lithium and with antipsychotic treatment [67]. Interestingly, in another study by Guloksuz et al. of 31 BD patients, lithium treatment affected the cytokine profile; namely, TNF-α and IL-4 levels in lithium monotherapy patients were higher than in both the medication-free BD patients and controls [68]. Remlinger-Molenda et al. carried out a more extended study of 121 BD patients, with 35 in immediate remission after mania, 41 in immediate remission after depression and 45 in sustained remission on lithium monotherapy or combined with other drugs. The control group was composed of 78 healthy individuals without psychiatric or immunological conditions. Patients in remission after mania showed higher concentrations of IL-10, while those in remission after depression exhibited higher levels of IFN-γ compared to healthy controls. However, cytokine concentrations in patients with sustained remission were similar to those of healthy subjects [69].

An ex vivo study showed similar results. Monocytes from non-lithium-treated BD patients had an abnormal IL-1β/IL-6 production ratio, with low IL-1β and high IL-6 levels that were restored by lithium treatment [47]. Ferensztajn-Rochowiak et al. [70] used lysed blood (apparently whole blood) to measure the mRNA expression of peripheral glial cell markers in 30 remitted BD patients consisting of 15 who had never been exposed to lithium [BD Li (-)] with a duration of illness of at least 10 years and of 15 patients treated continuously with lithium for 8–40 years [BD Li (+)] with a duration of illness of at least 15 years, as well as 15 healthy age- and sex-matched control subjects. In the BD Li (-) group, they observed increased levels of the transcription factors Olig1 and Olig 2 compared with the controls and increased glial fibrillary acidic protein (GFAP) levels compared with lithium-treated patients. In the BD Li (+) group, Olig 2 expression was higher than in the BD patients not taking lithium. The authors speculated that “*higher expression of peripheral mRNA markers in BD patients may involve ongoing inflammatory process, compensatory mechanisms and regenerative responses*” and that “*the beneficial effect of lithium may be related to its anti-inflammatory properties*”.

It is noteworthy that in bipolar patient’s gene expression studies reported a lack of difference in lymphocytes as compared with healthy controls [71] but also the presence of aberrantly expressed genes in monocytes [72]. Hence, the interpretation of the latter results is intricate. However, in lymphocytes and monocytes studies [71,72], lithium treatment induced significant alteration in cytokines expressions. Figure 1 illustrates a summary of the reviewed studies.

## 6. Conclusions

It is quite well established that inflammation/neuroinflammation is involved in the pathophysiology of at least one or some of the subgroups of BD and that lithium treatment is, among other effects, anti-inflammatory. However, two specific issues have not been thoroughly addressed: (1) Whether lithium exerts its anti-inflammatory/anti-neuroinflammatory effects in vivo in BD. (2) What the mechanism is by which lithium exerts its anti-inflammatory effects in the brain.

While much of the research on lithium’s anti-inflammatory characteristics has been and is conducted in the context of mental health disorders, there is growing interest in its potential utility for treating other inflammatory conditions. Conditions such as rheumatoid arthritis [73], inflammatory bowel disease [74] and even neuroinflammatory disorders like Alzheimer’s disease [75,76] are being explored as potential targets for lithium’s anti-inflammatory effects. However, it is important to note that lithium therapy requires careful monitoring due to its narrow therapeutic window and potential side effects, aiming at serum concentrations in the range of 0.5/0.6 to 1.1/1.2 mmol/L [77]. Common short-term side effects include nausea, tremors and cognitive dulling, while long-term use may lead to serious issues like chronic kidney disease, hypothyroidism and hyperparathyroidism, especially in women. Toxic lithium levels can cause severe symptoms, including organ failure [78]. Therefore, it is also crucial to determine the optimal dosing regimens for non-psychiatric applications.

Li influences the production of several pro- and anti-inflammatory cytokines. However, in studies utilizing different paradigms (in vitro in the cell culture of a variety of cells, in vivo in animal models, or in human subjects), the choice of different lithium concentrations and different agents to induce inflammation end up with different outcomes. In the clinical trials, the absence of information of confounding factors such as genetic variations and/or subtypes of bipolar disorder might explain the inconsistent results. Nevertheless, overall, lithium treatment appears to induce anti-inflammatory effects. As such, and as already suggested by others [79,80,81] and by us [82,83,84], it is reasonable to suggest that experiments of repurposing be carried out to investigate lithium’s use in other inflammatory disorders, such as neurodegenerative or autoimmune diseases.

To sum up, while research on lithium’s anti-inflammatory effects is still in its early stages, it holds promise as a potential therapeutic agent for a range of inflammatory conditions. Further investigation into its mechanisms of action and clinical efficacy is warranted to unravel its full potential in combating inflammation.

## Figures and Tables

**Figure 1 ijms-25-13277-f001:**
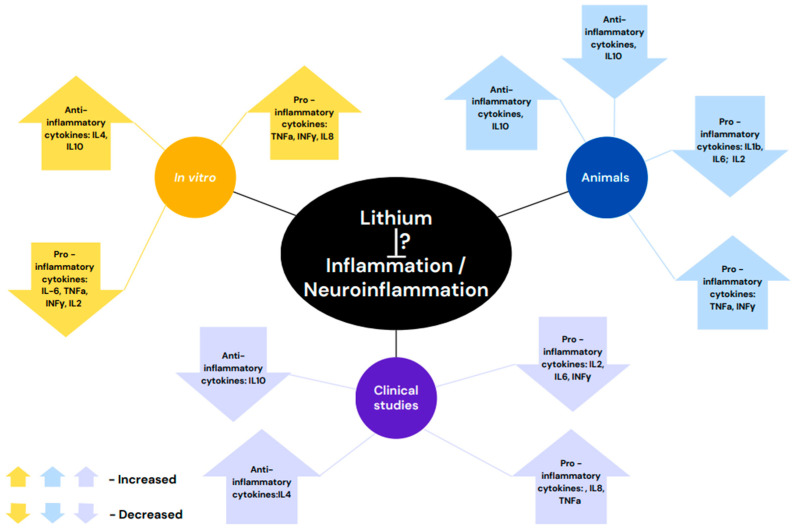
Lithium-induced effects on inflammation/neuroinflammation are not yet definite. This might be due to the variability in the methodology of the experiments. In vitro, animal and clinical studies report increased [43,54,58,64,68] anti-inflammatory cytokine levels. In animal models and in clinical trials decreased levels were also reported [27,60]. As for the levels of pro-inflammatory cytokines—either an increase [54,65,67,68] or a decrease [27,43,57,59,60,65] were found in all three paradigms.

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
