# Peer review of "Lithium, Inflammation and Neuroinflammation with Emphasis on Bipolar Disorder—A Narrative Review"

_ijms, 2024, doi:10.3390/ijms252413277_

Round 1

Reviewer 1 Report (Previous Reviewer 3)

Comments and Suggestions for Authors

Please format the references according to the journal's requirements. 

It would be a good idea to present shortly the methodology of your search. 

Author Response

Reviewer 1: (i) Please format the references according to the journal's requirements.

Response:    The references have been formatted according to the journal's requirements.

(ii) It would be a good idea to present shortly the methodology of your search.

Response:  A short description of the methodology of our search has now been included (lines 78-83).

Reviewer 2 Report (Previous Reviewer 2)

Comments and Suggestions for Authors

I am accepting the authors' reply and manuscript modification.

Author Response

Reviewer 2: I am accepting the authors' reply and manuscript modification.

Response: We thank the reviewer for this comment.

This manuscript is a resubmission of an earlier submission. The following is a list of the peer review reports and author responses from that submission.

Round 1

Reviewer 1 Report

Comments and Suggestions for Authors

I would like to begin by complimenting the authors for their review titled: "Lithium, Inflammation and Neuroinflammation with Emphasis on Bipolar Disorder." The analysis provided to examine the effects of lithium on conditions such as bipolar disorder and epilepsy across various fields, from in vitro studies to clinical trials, offers a clear overview on several topics. I would only suggest a few minor adjustments to the authors:

  • Some recent references could also be added, such as: doi: 10.1001/jamapsychiatry.2021.3170. doi: 10.1016/j.euroneuro.2021.10.003. DOI: 10.12740/PP/OnlineFirst/92740
  • I would recommend adding illustrative images in the different sections, as was done with Figure 1.
  • Additionally, I would suggest increasing the font size of the arrows in Figure 1, as they are currently unreadable.

Author Response

I would like to begin by complimenting the authors for their review titled: "Lithium, Inflammation and Neuroinflammation with Emphasis on Bipolar Disorder." The analysis provided to examine the effects of lithium on conditions such as bipolar disorder and epilepsy across various fields, from in vitro studies to clinical trials, offers a clear overview on several topics.

We thank the reviewer for these warm words.

I would only suggest a few minor adjustments to the authors:

  • Some recent references could also be added, such as: doi: 10.1001/jamapsychiatry. 2021.3170.  doi: 10.1016/j.euroneuro.2021.10.003. DOI: 10.12740/PP/OnlineFirst /92740

As suggested, we have now added these references to the references list, 4-7.

  • I would recommend adding illustrative images in the different sections, as was done with Figure 1.

Figure 1 already summarizes all the different sections in respect of lithium`s effects on anti- and pro- inflammatory cytokines. 

  • Additionally, I would suggest increasing the font size of the arrows in Figure 1, as they are currently unreadable.

As suggested, we have now increased the font size of the arrows in Figure 1.

Reviewer 2 Report

Comments and Suggestions for Authors

Review of “Lithium, Inflammation and Neuroinflammation with Emphasis on Bipolar Disorder”.

1. Weren't there papers on the effect of lithium on mRNA degradation? 

2. Lithium is very toxic, please describe lithium-based drug toxicity and provide a literature review - especially - long-term effects. Lithium is simply toxic to any cell: neuronal or immune cells.

3. Please also comment on gene-drug interactions, using the PharmGKB website and discuss particularly vulnerable patient populations: https://www.pharmgkb.org/search?query=lithium

3. There are plenty of anti-inflammatory and immunomodulatory drugs on the market. Why study lithium? It is so completely irrelevant.

4. All experiments, reviewed, either do not have proper controls (other anti-inflammatory drug comparator), or do not have control groups at all. In vitro cell experiments are never correlated with real biology, because in mammalian organisms the drugs are metabolized by the liver. Why use LPS or PHA, or other stimulants that are not used in humans? Please comment.

5. My concern is that the research on lithium and immunomodulation is pseudoscience, because it was proven only in vitro studies when you add the drug directly into already stressed-out and activated cells, that were taken out from their natural environment. 

Author Response

Review of “Lithium, Inflammation and Neuroinflammation with Emphasis on Bipolar Disorder”.

  1. Weren't there papers on the effect of lithium on mRNA degradation? 

Our literature search prior the preparation of the manuscript and following the reviewer’s remark failed to find studies reporting effects of lithium on degradation of inflammation-related mRNA.

  1. Lithium is very toxic, please describe lithium-based drug toxicity and provide a literature review - especially - long-term effects. Lithium is simply toxic to any cell: neuronal or immune cells.

At therapeutically relevant concentrations lithium is not toxic, neither to neuronal cells nor to other cells. Rather, it is neuroprotective and cell protective (see, for example, one of numerous papers, Neurosci Biobehav Rev. 2023;148:105148).

Lithium-based drug toxicity (following drastic overdose) is beyond the scope of this review.

3. Please also comment on gene-drug interactions, using the PharmGKB website and discuss particularly vulnerable patient populations: https://www.pharmgkb.org/ search?query=lithium

Gene-drug interactions are beyond the scope of this review.

  1. There are plenty of anti-inflammatory and immunomodulatory drugs on the market. Why study lithium? It is so completely irrelevant.

Lithium is the prototype drug for bipolar disorder. Given that neither the etiology of this disorder nor the mechanism of lithium’s beneficial effects are not yet unraveled, the present review, as Reviewer 1 writes “offers a clear overview on several topics” related to inflammation and neuroinflammation. This understanding may serve two goals: 1. If lithium`s effect on inflammation is involved in its mood stabilization action it may shed light on the pathophysiology of the disease. 2. Serve a rationale platform for the search/design of potential lithium mimetics/novel anti-inflammatory drugs. We have now added these implications to the Introduction p. 51-54. 

  1. All experiments, reviewed, either do not have proper controls (other anti-inflammatory drug comparator), or do not have control groups at all. In vitro cell experiments are never correlated with real biology, because in mammalian organisms the drugs are metabolized by the liver. Why use LPS or PHA, or other stimulants that are not used in humans? Please comment.

We absolutely disagree with the reviewer! e.g. the reviewer writes: “In vitro cell experiments are never correlated with real biology, because in mammalian organisms the drugs are metabolized by the liver.” Lithium, which is an ion, is not metabolized, it is just excreted by the kidneys. The reviewer also writes: “Why use LPS or PHA, or other stimulants that are not used in humans? LPS (lipopolysaccharide) is an outer membrane component of gram-negative bacteria. PHA (phytohemagglutinin) is a lectin, obtained from the red kidney bean which many human beings obtain through their diet. PHA binds T-cells membranes and stimulates metabolic activity.

Yet, to tackle the reviewer’s concern related to lithium’s toxicity, we added in the Conclusion section sentences concerning lithium’s therapeutic range levels and the potential hazards when higher concentrations are employed, lines 246-250.

  1. My concern is that the research on lithium and immunomodulation is pseudoscience, because it was proven only in vitro studies when you add the drug directly into already stressed-out and activated cells, that were taken out from their natural environment. 

We, again, strongly disagree with the reviewer! In sick people, in general, and in bipolar patients, in particular, multiple cells are “already stressed-out and activated cells”. It is only then that lithium treatment is started. Furthermore, in this manuscript we review lithium`s effects on inflammation in in vitro and in vivo studies, as well as in clinical trials, pointing out the differences among them.

Reviewer 3 Report

Comments and Suggestions for Authors

Although the topic might be of interest, the quality of the manuscript (in its current form) does not respect the journal's high standards (in my opinion). There are too many aspects that need to be improved for the manuscript to become suitable for publication. I highlight below the most relevant points to be considered and I suggest, after rewriting the article, to resubmit it.

1. What kind of paper is this? systematic review or narrative review? this should be stated for the title, in the abstract and in the introduction.

2. If you want to make a systematic review, it should respect the prisma guidelines. 

3. Abstract should be structured - introduction, materials and methods, results, conclusion (for both types of reviews). 

4. The introduction should be re-written. I suggest you use a 4-paragraph structure - epidemiology and importance of bipolar disorders, litium and its use in BD, role of inflammation in BD, link of litium to inflammation, aims of the article

5. A material and methods section should be added. 

6. Considering in vitro, animal, and clinical studies could seem a bit overwhelming at first. If you do a systematic review, I suggest you choose only one part (i.e. litium, neuroinflammation, and BD - perspective from clinical studies). 

Author Response

Although the topic might be of interest, the quality of the manuscript (in its current form) does not respect the journal's high standards (in my opinion). There are too many aspects that need to be improved for the manuscript to become suitable for publication. I highlight below the most relevant points to be considered and I suggest, after rewriting the article, to resubmit it.

  1. What kind of paper is this? systematic review or narrative review? this should be stated for the title, in the abstract and in the introduction.

Our manuscript is a narrative review. As suggested by the reviewer, this is now stated in the title, in the abstract and in the introduction. 

  1. If you want to make a systematic review, it should respect the prisma guidelines. 

Please see the response to point 1.

  1. Abstract should be structured - introduction, materials and methods, results, conclusion (for both types of reviews). 

Please see the response to point 1.

  1. The introduction should be re-written. I suggest you use a 4-paragraph structure - epidemiology and importance of bipolar disorders, litium and its use in BD, role of inflammation in BD, link of litium to inflammation, aims of the article
  2. A material and methods section should be added. 

According to the guidelines of the journal: "The structure can include an Abstract, Keywords, Introduction, Relevant Sections, Discussion, Conclusions, and Future Directions", a methods section is not a must.

  1. Considering in vitro, animal, and clinical studies could seem a bit overwhelming at first. If you do a systematic review, I suggest you choose only one part (i.e. litium, neuroinflammation, and BD - perspective from clinical studies).

Please see the response to point 1.

Reviewer 4 Report

Comments and Suggestions for Authors

First of all, I would like to congratulate the authors for addressing such a relevant and timely topic as the impact of lithium on inflammation and neuroinflammation, with a particular focus on bipolar disorder. This article provides a comprehensive and well-structured review of the effects of lithium in both in vitro studies and animal and clinical models, offering a thorough perspective on the therapeutic potential of this compound. The work stands out for its clear exposition and its ability to synthesize a large amount of information, making it easier to understand both the context and the relevance of the findings described. Additionally, the inclusion of studies covering various areas of the immune system and its interaction with lithium offers a broad and solid perspective on the topic, which will undoubtedly contribute significantly to future research in this field.

However, although the article is rigorous and valuable in many aspects, there are a few points that could be improved to further enhance the quality of the manuscript.

First, it would be advisable for the authors to delve deeper into the methodological limitations of some of the reviewed studies, especially those where lithium concentrations outside the therapeutic range are employed. While the effects observed at these concentrations are mentioned, including a more detailed discussion on the clinical relevance of these results would help to better contextualize the applicability of the findings. This clarification would be particularly useful in the in vitro studies, where the concentrations used often do not reflect the physiological conditions of lithium treatment in patients.

Furthermore, although the article addresses the differences between in vitro studies and clinical studies in patients with bipolar disorder, it would be interesting for the authors to expand the discussion on the possible reasons behind these discrepancies. A deeper reflection on the mechanisms underlying the differences between experimental models and clinical results could offer a more critical and detailed view of lithium’s impact on inflammation and neuroinflammation. This would not only enrich the article but also provide a better understanding of the difficulties and opportunities when using lithium as a therapeutic agent in different patient populations.

It is also suggested that the authors consider including comparative graphs or data visualizations, particularly in the sections where the concentrations of pro-inflammatory and anti-inflammatory cytokines are discussed. Visual presentation of these key results would not only facilitate understanding of the data but also provide greater clarity regarding the trends observed in the different reviewed studies. This improvement would make the article more accessible and appealing, especially for those unfamiliar with the detailed analysis of these biomarkers.

Another aspect that could be strengthened is the discussion on individual variations in response to lithium, especially concerning genetic factors and subtypes of bipolar disorder. It would be useful for the authors to explore how these individual differences could influence the effectiveness of lithium as an anti-inflammatory treatment and to what extent these variations should be considered when interpreting study results. Including this reflection would add an additional layer of depth to the article's analysis, emphasizing the complexity of personalized treatment in managing bipolar disorder.

Regarding the conclusions, a more focused summary of the key findings and the gaps in current knowledge would better highlight the article's contributions and strengthen its impact. Furthermore, the final section could benefit from greater speculation on the therapeutic prospects of lithium in other inflammatory disorders, such as neurodegenerative or autoimmune diseases. Although this possibility is briefly mentioned in the manuscript, a more detailed analysis of lithium’s potential applications in these fields could open new avenues for research and expand the scope of the article.

Author Response

First of all, I would like to congratulate the authors for addressing such a relevant and timely topic as the impact of lithium on inflammation and neuroinflammation, with a particular focus on bipolar disorder. This article provides a comprehensive and well-structured review of the effects of lithium in both in vitro studies and animal and clinical models, offering a thorough perspective on the therapeutic potential of this compound. The work stands out for its clear exposition and its ability to synthesize a large amount of information, making it easier to understand both the context and the relevance of the findings described. Additionally, the inclusion of studies covering various areas of the immune system and its interaction with lithium offers a broad and solid perspective on the topic, which will undoubtedly contribute significantly to future research in this field.

Response: We thank the reviewer for this supportive comment.

However, although the article is rigorous and valuable in many aspects, there are a few points that could be improved to further enhance the quality of the manuscript.

First, it would be advisable for the authors to delve deeper into the methodological limitations of some of the reviewed studies, especially those where lithium concentrations outside the therapeutic range are employed. While the effects observed at these concentrations are mentioned, including a more detailed discussion on the clinical relevance of these results would help to better contextualize the applicability of the findings. This clarification would be particularly useful in the in vitro studies, where the concentrations used often do not reflect the physiological conditions of lithium treatment in patients.

Response: As suggested by the reviewer we have now added a sentence (lines 77-79) commenting on the methodological limitations of in vitro  studies where lithium concentrations employed were outside the therapeutic range. In addition, we are now mentioning lithium`s concentration used in the in vitro studies cited, particularly if they exceed the therapeutic range.

Furthermore, although the article addresses the differences between in vitro studies and clinical studies in patients with bipolar disorder, it would be interesting for the authors to expand the discussion on the possible reasons behind these discrepancies. A deeper reflection on the mechanisms underlying the differences between experimental models and clinical results could offer a more critical and detailed view of lithium’s impact on inflammation and neuroinflammation. This would not only enrich the article but also provide a better understanding of the difficulties and opportunities when using lithium as a therapeutic agent in different patient populations.

Response: We added in the conclusion section the differences between in vitro, in vivo and etc. lines: 255-259.

It is also suggested that the authors consider including comparative graphs or data visualizations, particularly in the sections where the concentrations of pro-inflammatory and anti-inflammatory cytokines are discussed. Visual presentation of these key results would not only facilitate understanding of the data but also provide greater clarity regarding the trends observed in the different reviewed studies. This improvement would make the article more accessible and appealing, especially for those unfamiliar with the detailed analysis of these biomarkers.

Response:  The reviewer’s attention is drawn to the fact that Figure 1 provides a visual presentation summarizing the results of the effects of lithium on pro- and anti-inflammatory cytokines in in vitro, in vivo and clinical trials.

Another aspect that could be strengthened is the discussion on individual variations in response to lithium, especially concerning genetic factors and subtypes of bipolar disorder. It would be useful for the authors to explore how these individual differences could influence the effectiveness of lithium as an anti-inflammatory treatment and to what extent these variations should be considered when interpreting study results. Including this reflection would add an additional layer of depth to the article's analysis, emphasizing the complexity of personalized treatment in managing bipolar disorder.

Response: We thank the reviewer for this comment. Carrying out a thorough literature search we could not allocate studies concerning variations in genetics factors or subtypes of bipolar disorder related to inflammation. We now discuss this issue under Conclusions, lines 256-258.

Regarding the conclusions, a more focused summary of the key findings and the gaps in current knowledge would better highlight the article's contributions and strengthen its impact. Furthermore, the final section could benefit from greater speculation on the therapeutic prospects of lithium in other inflammatory disorders, such as neurodegenerative or autoimmune diseases. Although this possibility is briefly mentioned in the manuscript, a more detailed analysis of lithium’s potential applications in these fields could open new avenues for research and expand the scope of the article.

Response: As suggested by the reviewer, we added under the Conclusion section an overview of the gaps in current knowledge and their possible causes, lines 253-258, and speculation on the therapeutic prospects of lithium in other inflammatory disorders, such as neurodegenerative or autoimmune diseases, lines 258-262.